# Fighting the Consequences of the COVID-19 Pandemic: Mindfulness, Exercise, and Nutrition Practices to Reduce Eating Disorders and Promote Sustainability

Sara Baldassano [1] , Anna Alioto [2], Alessandra Amato [2] , Carlo Rossi [2] , Giulia Messina [2],
Maria Roberta Bruno [2] , Roberta Stallone [2] and Patrizia Proia [2,*]

[1]    Department of Biological Chemical and Pharmaceutical Sciences and Technologies (STEBICEF), University of Palermo, 90128 Palermo, Italy

[2]    Sport and Exercise Sciences Research Unit, Department of Psychology, Educational Science and Human Movement, University of Palermo, 90100 Palermo, Italy

*    Correspondence: patrizia.proia@unipa.it

**Abstract:** Over the past two years, the world's population has been tested by the COVID-19 health emergency. This has changed population habits worldwide by encouraging a sedentary lifestyle and overnutrition. Isolation and reduction of social life, for most of the population, was mandatory but it quickly became a new lifestyle. Nowadays, we are encountering the consequences with an increase in nutritional associated disorders and conditions that cause illnesses in the general population. These disorders include diet excesses that lead to obesity and diet deficiencies and malnutrition which could rapidly lead to death. These eating disorders are very complex to manage because they become mental disorders which can negatively impact physical or mental health. This work will disucss the benefits associated with the mindfulness–exercise–nutrition (MEN) technique. From a nutritional point of view it will focus on the nutritional effect of a plant-based diet, such as the Mediterranean diet (MD) which has a high tryptophan content which can increase serotonin (the "feel good" hormone) levels. The MEN technique takes a multidisciplinary approach and aims to integrate healthy behaviors into clinical practice using healthy eating, active living, and mindfulness. This method includes controlled physical movements, stretching techniques such as yoga, and aerobic exercise to achieve optimal mental and physical health. This literature review, carried out using the PubMed, ScienceDirect, Scopus, and Google Scholar databases, aims to investigate the latest research on this topic. This study may be useful for healthcare professionals and clinicians and may help patients to be more self-aware, encouraging them to lead a healthier lifestyle, make thoughtful choices, and ameliorate their mental health. The final aim of this study is to promote physiological homeostasis and well-being.

**Keywords:** tryptophan's shunt; malnutrition; obesity; plant-based diet; Mediterranean diet; homeostasis; healthy eating; psychological disorders

## 1. Introduction

There has been increased research interest in lifestyle patterns [1–3] and the possible relationship that they have with psychological disorders. This relationship has been explored in several studies which have shown that diet and sedentariness are among the possible factors that can contribute to the establishment of some disorders [4]. The recent pandemic has been a clear example of how significant lifestyle disruptions can cause changes in social and eating habits and significantly affect mood and physical functioning [5,6]. Among the manifest of psychological consequences, anxiety and depression are those that have most threatened the world's population during the pandemic health emergency [7]. In light of these findings across the literature, our study aims to correlate the positive effect of mindfulness, aerobic exercise, and diet with psychological disorders with the final aim

of promoting sustainability. In fact, promoting mindfulness practice can contribute to more sustainable ways of life and to greater well-being. This includes the promotion of a sustainable plant diet.

Physical and mental well-being may be promoted not only for individuals with psychological disorders such as anxiety and depression, but also for those who have suffered stress as a result of the global pandemic.

## 2. Materials and Methods

This narrative review was conducted using a series of databases to retrieve relevant articles: MEDLINE (PubMed), ScienceDirect, Web of Science, Scopus, and Google Scholar. A flow diagram illustrating the various steps of article selection is shown in Figure 1. The search terms we used were: ("Physical activity" or "Exercise" and "Mindfulness")/("Nutrition" or "Diet" and "Tryptophan" or "Mediterranean Diet")/("Gut–brain" or "GBA" and "Serotonin" or "Exercise" and "Gut"). In addition, we filtered studies based on the following: if there was a full text available, if studies were on humans, and if they were available in English. We decided to perform a narrative review because it represents a good way to provide an overview of the existing literature on a specific topic. Unlike systematic reviews or meta-analyses, which identify and analyze existing studies, a narrative review generally describes research findings on a topic in a more explanatory way. This makes it possible to explain how a specific topic has evolved and is studied over time, to discuss trends that have emerged in the research, and to identify the main areas knowledge in the existing literature.

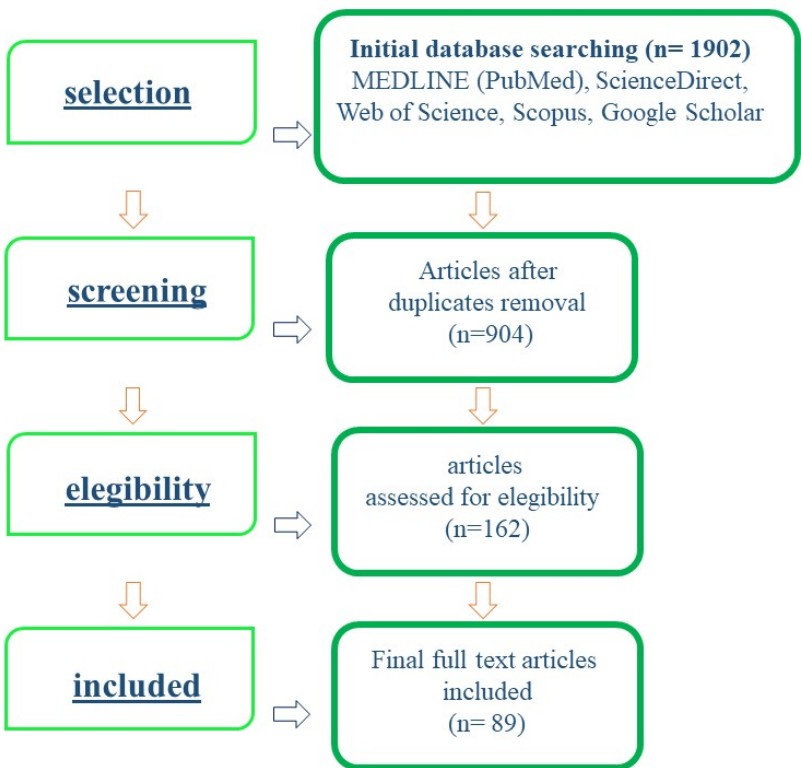

**Figure 1.** Flow diagram of the study selection process and the eligibility screening.

The studies were further selected based on a series of inclusion and exclusion criteria. The inclusion criteria were: (a) only peer-reviewed articles written in English, (b) randomized and nonrandomized studies, (c) observational and pilot studies published from 2002 to 2022 (d) meta-analysis, and (e) reviews. There were some exceptions to (c), with some pioneering studies before 2002 included as they were highly relevant to the aims of our study. The exclusion criteria were: (a) unpublished studies, (b) editorials, (c) letters to the

editor, (d) essays, (e) conference proceedings, and (f) abstracts were not included. Studies based on (g) children, (h) pregnant women, (i) in vitro experiments were also excluded.

The included literature was not limited based upon the frequency, intensity, type, or time of exercise performed, or other variables such as gender, age, sample size, or duration of the follow-up. However, we limited retrieved papers to those that were studies on healthy individuals.

We then used the literature to draw conclusions as to how the MEN technique can lead to favorable outcomes for those with anxiety and depression, or pandemic-related stress.

## 3. Impact of Nutrition in Systemic Homeostasis and Human Well-Being

A study performed before the pandemic by [8] aimed at identifying a metabolic biomarker for a more accurate diagnosis of the possible comorbidity of both anxiety and depression. In their study, the researchers tried to identify different metabolites present in the urine in a group of 32 patients screened for depression and anxiety disorders following the criteria indicated in the diagnostic and statistical manual of mental disorders (DMS) [8,9].

Among the metabolites found at the urinary level, four were significantly present in the patient group, relative to the control group. Among those was *N*-methylnicotinamide, a final product of the metabolism of nicotinamide, which in turn is bound to nicotinic acid and to tryptophan (TRP), a biochemical precursor of serotonin. The presence of this biomarker at the urinary level was considered an indication of abnormalities in the biosynthesis of serotonin, the alterations of which are implicated in the etiopathology of anxiety and depressive disorders [8]. Thus, with TRP being the biochemical precursor of both nicotinic acid and serotonin, altered levels of downstream metabolites in nicotinic acid metabolism could indicate disturbance of serotonin biosynthesis in patients with depression and anxiety disorders [10].

Accordingly, the discovery of TRP metabolites in the group of biomarkers which may cause anxiety and depressive disorders might suggest an involvement of these biomarkers in psychological, and consequently physical well-being. Abnormalities in this metabolism also involve an alteration in the kynurenine pathway (KP) and nicotinamide metabolism through the production of nicotinamide adenine dinucleotide ($NAD^+$), as well as the biosynthesis of 5-Hydroxytryptamine (5-HT). As 5-HT is a precursor to serotonin, alterations of 5-HT may affect the serotonergic system, which is a major pathway in which psychiatric disorders may develop [11,12].

While disorders in metabolites can lead to pathological circumstances related to well-being, behavioral changes, such as making proactive changes to lifestyle, are also important to consider [13–15]. A way in which pathology may be counteracted is through the practice of mindfulness, exercise, and healthy eating, which we define using the acronym "MEN". The practice of "Mindfulness–Exercise–Nutrition" (MEN) aims to ameliorate anxiety and depression through a state of psychophysical well-being [13]. Although there is little scientific literature referring to MEN as an integrated approach, there is one study that examined its use in breast-cancer survivors. However, the study did not include a biological link that explains the benefits [16] or show the biological effect alone, and it did not include any specific guidelines about diet and exercise [17]. In terms of diet, we acknowledge the MD as being defined as a sustainable diet. In 2010, the FAO in collaboration with Biodiversity International reached a scientific position on what is defined as a sustainable diet: "Sustainable diets are those diets with low environmental impact which contribute to food and nutrition security and healthy life for generations present and future. Sustainable diets are protective and respectful of biodiversity and ecosystems, culturally acceptable, accessible, economically equitable and convenient; nutritionally adequate, safe and wholesome; optimizing natural and human resources" [18]. Diet sustainability is the contribution that diet makes at a sustainable level and this is not only an important goal, but also an essential means to achieve a sustainable food system [19]. The Mediterranean diet (MD) is often used as an example of a sustainable diet [18,20] that promotes a sustainable food

system [21,22]. The MD is a dietary pattern rich in grains, fruits, vegetables, legumes, nuts, seeds, and olives, with olive oil as the major source of added fats, along with a high to moderate intake of seafood and fruits, a moderate consumption of eggs, poultry, and dairy products (cheese and yoghurt), a low consumption of red meat and a moderate consumption of alcohol (mainly wine during meals) [23,24]. A Mediterranean diet is able to aid in the prevention of the development of cardiovascular diseases of those who adopt it not only in populations living in the Mediterranean area [25–27], and can reduce the risk of diabetes [28] and metabolic syndromes [29,30]. Furthermore, a positive association has been demonstrated between a MD and the risk of various types of cancer [31], depression, and cognitive impairment [32], and it has beneficial effects on sleep quality in the adult population [33]. However, it should be emphasized that mindfulness correlates with higher levels of moderate and vigorous physical activity, fruit and vegetable intake, and lower levels of saturated fat, and can contribute to a state of well-being even in healthy subjects, as suggested in a study by Gilbert et al.

## 4. Psychological Disorders: Anxiety and Depression

According to the diagnostic and statistical manual of mental disorders (DSM-5) [9], anxiety and depression disorders represent pathological states that affect the behavioral, relational, cognitive (thinking, ideation, concentration, attention, and problem-solving skills), or affective (mood, emotions, feelings, and anxiety) spheres of a person [34]. This can lead to a range of resultant social and health issues, such as detrimental effects on family life, work, eating habits, and sleep quality, with further repercussions on physical health, lifestyle, and quality of life [34].

To date, the origin of anxiety and depressive disorders are not completely understood. However, multiple literature sources suggest that there is a multifactorial etiopathogenesis: genetic background, environmental factors, stressful events, or drug abuse [35–37]. As anxiety and depression frequently occur comorbidly, the previous edition of DSM-4 [38] listed a mixed anxiety–depressive disorder, which is characterized by a dysphoric mood, lasting for at least a month, associated with at least four of the various distinctive signs such as difficulty concentrating, fatigue, exhaustion, irritability, despair, ease in crying, negative worries, and predictions, poor sleep quality, low self-esteem followed by contempt for oneself [9].

The rapid spread of coronavirus (COVID-19), which led to a pandemic over the last two years, has also led to an increased focus on mental health disorders. This is because measures applied to mitigate the spread of the SARS-CoV-2 virus (such as quarantine and social isolation) may have contributed to the onset of long-term psychological distress, including high levels of depression, stress, and anxiety [39].

Given the recent pandemic and related psychosocial health issues, it is imperative that strategies are designed to protect the psychological well-being of people. This could help provide concrete suggestions on how to prevent nascent psychological problems from evolving into serious long-term mental health conditions [40,41].

The WHO has reported a summary of data related to mental health from the Global Burden of Disease 2020 which is a concern. They reported that the global prevalence of all psychological disorders has increased by 25% compared to the pre-health-emergency period [42].

These worrying data have prompted the WHO to leverage the countries interviewed by inviting them to implement measures for the protection of mental health and psychosocial support. However, the WHO itself reports that there are still major gaps concerning the prevention of these diseases as well as the promotion of psychological well-being through interventions based on awareness and the understanding of the warning signs of such diseases [42].

## 5. Psychobiological Biomarkers

### 5.1. Serotonin Pathway

Serotonin (5–hydroxytryptamine or 5–HT), widely known as the "good mood" hormone, is a monoamine, a derivative of the essential amino acid TRP obtained from food sources through the action of the tryptophan hydroxylase enzyme (TPH) [43]. It is a well-known neurotransmitter that acts at the level of the central nervous system (CNS) and it is a blood factor and a neurohormone that controls the function of several peripheral organs [44]. It also acts as a modulator of important physiological processes such as appetite, sleep, and mood, thus becoming a marker for the treatment of psychological disorders such as anxiety and depression [45]. The majority of the body's serotonin (95%) is produced in the intestine, functioning as a hormone and autocrine, paracrine, and endocrine signaling.

It is of utmost importance that the intestine receives regulatory signals from the CNS for proper functioning. Recently, some studies have proved that the intestine is also capable of transmitting signals to the CNS which are then received by the brain. This two-way communication between the intestine and the brain is called "gut–brain axis" [46]. Dysfunctional intestinal microbiomes may be associated with behavioral disorders such as depression. Additionally, for this reason, serotonin, or its precursor TRP, seems to be an important molecule [47].

To further understand the physiologically relevant functions of 5-HT receptors, it is important to understand their distribution [48]. Several receptors are located in the CNS as well as the gut, so studying these receptors could lead to an increased understanding of brain–gut interactions [49]. Currently, there have been 14 different receptor subtypes belonging to the seven main receptor classes (5–HT1–7), which have been described in different parts of the brain, however their functions are still under investigation [50].

Many studies have investigated the different response of these receptors, but serotonin (5-HT) has a stimulatory effect on the hypothalamic–pituitary–adrenal (HPA) axis in both humans and rodents [51]. This interaction is mediated by the 5-HT1A receptor. However, only male rodents and not humans, respond to 5-HT1A antagonism to show increased corticosterone responses to stress [51,52].

The correlation between circulating levels of peripheral 5-HT and metabolic diseases has been discovered. This has been found to be attributable to the enzyme TPH, which in mammals catalyzed the 5-HT synthesis from TRP [53,54]. The enzyme has two isoforms: *Tph 1*, expressed mainly by enterochromaffin cells and adipocytes, and *Tph2* expressed in neurons of the nuclei of the raphe, in the brain stem and in the enteric nervous system (ENS). Therefore, 5-HT has central and peripheral pools that operate separately.

Recently, Jones et al. discovered a modern function of 5-HT, highlighting an emerging pathway by which gut microbiota influences host health [55]. Since 5-HT cannot cross the blood–brain barrier, the central and peripheral pools of 5-HT are physically separate and act differently. In the brain, central 5-HT has an anorexigenic effect, whilst peripheral 5-HT acts on energy homeostasis. This action peripherally occurs in areas such as the liver and adipose tissue, in which the increase of 5-HT is associated with obesity and the development of metabolic diseases [56]. Enterochromaffin (EC) cells act as sensors within the intestinal lumen and are affected by the intestinal microbiome. In particular, the interactions between the intestinal microbiota and EC cells are important for glucose homeostasis. This mechanism may target the gut–brain axis signaling through gut microbiota and gut-derived 5-HT interaction.

The reduction of intestinal 5-HT levels has been shown to protect against diet-induced obesity, fatty liver disease, and glucose intolerance [57,58]. Except in cases where it is necessary to use drug therapy, the literature has provided evidence that suggests promoting the secretion of 5-HT could help counteract these psychological disorders.

The biochemical mechanism that can promote the synthesis of 5-HT after a period of physical activity has not yet been studied. Exercise represents one of the most common approaches for health and wellness, particularly forms of physical activity (PA) defined as mindful movement. Such exercises include yoga and Tai Chi and have been used increas-

ingly as complementary health approaches. These are considered to be nonpharmacological and unconventional methods, validated for the treatment of anxiety and depression disorders [59,60]. For example, exercises such as meditation, yoga, breathing exercises, Tai Chi, or Qi Gong have provided statistically significant (but variable), results in terms of effectiveness, depending on style, duration over time, intensity, and if performed in combination [59,60].

Mindfulness-based stress reduction currently seems to be a widespread method which has yielded favorable results for well-being. Scientific evidence has shown that eight weeks of meditative practices are effective for improving emotional stability, the perception of well-being, and reducing the symptoms of stress, anxiety, and depression [61]. It also increases the immune response, improves blood pressure, and regulates the frequency of breathing, raising dopamine values and reducing GABA, adrenaline, and cortisol which are associated with anxious states. Unfortunately, the mechanisms that correlate physiological biomarkers and the effect of mindfulness remain poorly understood [62,63].

Despite this lack of understanding, several studies have investigated epigenetic changes as a result of mindfulness practices. For example, the methylation of the transporter gene of 5-HT, SLC6A4, which catalyzes the reuptake of 5-HT within the synaptic cleft [64–66] has been reported. In fact, researchers found that a significant decrease in methylation was observed at a genomic site of this gene after a period of 12 weeks with a mindfulness program, suggesting an effective practice in the treatment of stress-related disorders. Another study has shown that methylation in a CpG island (correlated with silencing of genes) in the 5' region of the 5HTT gene, has been associated with a decrease in gene expression which, under stressful conditions, would induce changes in behavioral response [64]. Moreover, the postintervention mindfulness-program (MP) has shown that it can contribute to the reduction of gene methylation with a "preventive" effect on the onset of psychological stress [17].

*5.2. Kynurenine's Pathway*

Beneficial effects on mental health may also arise due to muscle pathways that are stimulated through exercise. In a recent study by Valente-Silva et al., the authors reported an interesting discovery of a biochemical pathway in skeletal muscle through which kynurenine (Kyn) [66] and TRP could have a positive impact on mental health. This has found to be controlled by the peroxisome proliferator-activated receptor $\gamma$ coactivator 1$\alpha$ (PGC-1 $\alpha$) involved in the cellular adaptive process [66]. The benefits of exercise in people with depression are recognized, although the underlying mechanisms remain unknown [67,68]. Exercise induces the production of PGC-1$\alpha$1 in skeletal muscle and kynurenine aminotransferases (KATs) enzymes in turn convert kynurenine into kynurenic acid, which can act as a protective mechanism from depression [69].

Agudelo et al. described a mechanism by which the PGC-1$\alpha$1 factor, produced in skeletal muscle, and whose synthesis is induced by training, changes the metabolism of kynurenine and protects against stress-induced depression. The activation of the pathway followed by PGC-1$\alpha$1 increased the muscle expression of some KATs, thus improving the conversion of kynurenine into kynurenic acid, a metabolite unable to cross the blood–brain barrier [70,71]. Reduction in plasma kynurenine levels has been shown to protect the brain from the stress-induced changes associated with depression [69].

In order for 5-HT synthesis to occur adequately throughout the body and the CNS, there must be sufficient intake of the essential amino acid TPH, through nutrition [72], which promotes psychological well-being [73]. Diet therefore plays an important role in enhancing 5-HT production and thus potentially protecting against the effects of conditions such as depression. For example, a direct association has been found between the consumption of processed foods (e.g., processed meats and refined cereals) and a greater risk of developing depression [74]. However, the MD and other more traditional dietary patterns, rich in fruits, vegetables, fish, and whole grains, have been correlated with a lower rate of depression [75].

## 6. Mediterranean Diet and Environmental Sustainability

Many studies have shown that a healthy lifestyle is associated with the prevention of psychological disorders [62,75]. These include nutrition and regular physical activity. While the field of nutrition remains a very complex field, the MD, also known as the "good mood diet", stands out among alternative dietary models [76,77]. It was described for the first time, in the 1950s, by the American researcher Ancel Keys, following the observation of low mortality rates due to coronary heart diseases, usually correlated with what are now called well-being diseases (e.g., obesity, diabetes, hypertension), in some populations that lived in the Mediterranean areas, such as Greeks and southern Italians [25,78]. The MD has several advantages in terms of sustainability. Considering the three dimensions of sustainability (environmental, social, and economic), it has low environmental impacts, is characterized by high sociocultural food values, and allows positive local economic returns. It has been widely demonstrated that various health benefits can be linked to the adoption of a food model inspired by the MD [79–93].

Despite all these positive factors, we find some situations, such as the aggressive cultivation techniques in Spain, are detrimental to sustainability in environmental terms, and lead to an increase in temperature in Spain.

Since the 1950s, the interest in the MD pattern has been extensively studied and approved by the medical and scientific community as a positive dietary approach in terms of food quality and a better quality of life. It is characterized by the high consumption of plant foods (fruits and vegetables), whole grains (traditionally minimally refined), olive oil (used as the principal source of fat), dried fruits (such as nuts and seeds), seafood, seasonally fresh and locally grown foods, a moderate consumption of dairy products, poultry, and red wine generally with meals, and a low consumption of red meats and saturated lipids.

The components of the MD (e.g., extra-virgin olive oil and nuts) have well documented health benefits [94,95]. A meta-analysis performed from Psaltopoulou et al., examined the association between adherence to a MD style and depression [32]. Tryptophan is amongst the micronutrients contained in MDs, is necessary for human nutrition, and has positive systemic effects, such as on bone density [3]. It also has desirable effects on metabolism and physiological functions and is a precursor to important molecules such as serotonin and melatonin [96]. The minimum daily requirement is 250 mg for men and 150 mg for women [97]. Plasma TPH levels are determined by a balance between dietary intake [98] and its removal from plasma plays an essential role in protein synthesis. An alteration of this balance decreases the reserves of serotonin in the brain, which can lead to the onset of depression or behavioral disturbances [99]. Serotonin is a key element of the brain–gut axis previously discussed, that links the emotional and cognitive centers of the brain with peripheral control and function of the gut [2]. As the MD has a high content of the amino acid TPH, the diet has been seen to have a positive impact on sleep quality and can protect against anxiety and stress.

## 7. Mindfulness and Physical Activity

The health emergency of COVID-19 has imposed drastic measures in the daily life of the world's population, leading to psychological problems, including anxiety and depression, due to social isolation [100]. Serotonin (5-HT) and its precursor TRP if imbalanced, can lead to psychological perturbations. In the context of depression, TRP can be metabolized in the serotonin and kynurenine pathways [101] which are involved in healthy and pathological states. Furthermore, these products can mediate the effects of exercise, mood, and neuronal excitability and, ultimately, communicate with the microbiota of the gut [102].

In the treatment of anxiety and depression disorders, many have opted for complementary and integrative therapies, such as mindfulness practices (meditation, yoga, Tai Chi, Qi Gong) and exercise, which have been demonstrated to have favorable effects on mental health [60]. Mindfulness draws its origins from Buddhist psychology and, according to Brown et al., can be defined as a form of meditation that invites people to focus attention

on the present moment, without judgment [103]. The benefits of mindfulness are within the emotional, interpersonal, and intrapersonal dimensions [104]. Additionally, physical exercise is widely recognized as a preventive factor which can also help with disease management and improves mental health, improving resilience, quality of life, and well-being in general [1–3,105].

Combined mindfulness and exercise practices are some of the activities suggested for the nonpharmacological treatment of mental disorders. Mindfulness is an approach focused on the mind–body connection, and is characterized by exercises that include controlled physical movements, stretching of the whole body, breathing techniques, and a component of meditation, such as yoga [106]. Several studies have investigated the effectiveness of yoga to counteract the symptoms of depression. However, the protocols to be followed in terms of duration and frequency, remain undefined. For example, a meta-analysis by Bridges et al. examined exercise parameters for depression [107]. The intervention periods ranged from 1 to 24 weeks, with a frequency of once to every day a week, with the duration per session from 12 to 90 min [107]. The study indicated that there are no differences in terms of reduction of depression symptoms when yoga was practiced once compared to twice a week. However, more frequent sessions were associated with reductions in anxiety symptoms. The duration in most reports was three to 24 weeks, with frequencies ranging from once a week to every day for 40–100 min per session. On average it seems that, to have a positive effect, it is necessary to practice mindfulness for a period of more than eight weeks.

In summary, yoga may be suggested as therapy for depression, but it is preferred as an additional treatment for depression and anxiety disorders [60]. Similarly, a large body of evidence supports exercise as a "medicine" as it has preventive and therapeutic effects on various pathologies. In recent times, the effects of exercise on mental conditions has also been considered. However, it may be difficult to determine the precise exercise parameters that can be used to treat illnesses. For example, a meta-analysis of randomized controlled trials demonstrated the difficulty in selecting the type of exercise suitable for depression [106].

With regard to physical exercise, aerobic or resistance training, in particular, seem to have a greater effect on the metabolism of TRP, although this may depend not only on the individual's physical fitness but also on the crosstalk with other nutrients. It has recently been demonstrated that training can cause an increase in the expression of the genes that codify for KATs enzymes in skeletal muscle, thus shifting the peripheral metabolism of kynurenine towards the production of kynurenic acid [108]. Consequently, the reduction of kynurenine accumulation in the central nervous system results in a positive contribution to mental health and can reduce stress-induced depressive symptoms [102]. For example, during aerobic exercise there is an increase in circulating free fatty acids which displace TRP from albumin, increasing the levels of free TRP that are absorbed and metabolized, with a direct impact on the KP. The increase in TRP uptake also increases its metabolites in various regions of the brain that are associated with the control of mood and fatigue [109].

It has also been shown that skeletal muscle responds to the training stimulus of physical exercise, especially aerobic exercise, by increasing the expression of PGC-1$\alpha$ coactivators [110]. In a study by Pietta-Dias, participants trained two times a week using a treadmill at different intensities which were defined relative to their maximum heart rate (HRmax) [111]. The study participants followed three different mesocycles of walking: the first for 20 min at 60–65% of their HRmax, the second for 30 min at 70–75% of their HRmax, and the last for 35 min at 80–85% of their HRmax [111].

Taken together, the findings from the literature seem to suggest that mindfulness combined with endurance exercise can cause muscle expression of KATs and the detoxification of kyn accumulation in the brain [66]. This can lead to alterations associated with stress-induced depression, thereby offering potential therapeutic targets of antidepressant drugs [66].

## 8. Discussion

The scientific evidence reported in our study is an attempt to confirm the psychophysical benefits that come from the practice of mindfulness, exercise, and healthy eating [67,72,102]. The common thread between MEN and psychological well-being, and consequently, physical well-being, also seems to depend on the metabolism of TRP (monoamine 5-HT synthesis or the kynurenine pathway), due to its role in CNS regulation [62]. Physical exercise increases the synthesis of the transcriptional coactivator PGC-1α1 in skeletal muscle, controlling many of the adaptations to physical activity and inducing the expression of KATs enzymes. In turn, this shifts peripheral kynurenine to kynurenic acid and counteracts the toxic effects of kynurenine accumulation. This, added to the increase of TPH activity induced by physical activity, increases production of 5-HT and consequently serotonin. The combination of exercise with mindfulness practice, based on meditation, gentle movements, and breathing techniques which focus on the mind–body connection, can raise the awareness of oneself and the state of relaxation. Yoga and Tai Chi are some examples of mindful exercise.

The plasma concentration of TRP is influenced by many nutritional, physiological, and circulant factors. From a nutritional point of view, the MD has high TPH content food which increases the circulating TPH level. Additionally, it has been demonstrated to have health benefits and is correlated with the reduction of depression through the increase of serotonin and melatonin production [72,109].

Many studies have found that the MD is a sustainable option [79–84,112]. Further, the MD appears to be the most environmentally friendly diet [79,81,85–87,112]. It is a dietary pattern characterized by a moderate consumption of eggs, poultry, and dairy products (cheese and yoghurt) and a low consumption of red meat [88–91]. It is now known that the MD has had a lower environmental impact than Western diets with production ranging between 0.9 and 6.88 kg of $CO_2$/day per capita, and a reduced water consumption ranging between 600 and 5280 $m^3$/day per capita [92]. Furthermore, the MD has been recognized by UNESCO as an intangible cultural heritage.

Despite its growing popularity around the world, adherence to the MD is declining due to multifactorial influences such as globalization, population growth, and socioeconomic changes.

In summary, it is possible to suggest a practical approach to prevent and counteract psychological distress effectively reducing the manifestation of long-term disorders such as depression or anxiety.

Based on the studies reported, we suggest an intervention period of more than eight weeks which follows the guidelines below:

1. Follow a MD; Table 1 lists the dietary suggestions [93];
2. Perform mindfulness practices such as yoga or Tai Chi three times a week for a duration of no less than 60 min;
3. Perform physical activity using a treadmill two or three times a week (even at home). If we divide the 24 weeks into three mesocycles, in the first mesocycle, it is better to start with 20 min of total exercise at 60–65% of the HRmax, which can be calculated using the formula from Karvonen et al. [113]. In the second mesocycle it is possible to increase to 30 min at 70–75% of the HRmax, and in the last mesocycle it can be increased to 35 min at 80–85% of the HRmax.

**Table 1.** Information on typical foods of the Mediterranean diet and their frequency of consumption.

| Food in a Mediterranean Diet | Serving |
| --- | --- |
| **Read and processed meat** (beef, pork) <br> **Sweets** (cakes, sugar, honey, sugar drinks) | Very low consumption |
| **Eggs** <br> **Potatoes** <br> **Poultry** <br> **Seafood** <br> (Red mullet, sardines, anchovies, swordfish, sea bream, sea bass, squid, octopus) | Moderate consumption |
| **Dairy products** <br> (fresh milk, low-fat yogurt) | Low consumption |
| **Olive oil** <br> (virgin and extra virgin olive oil) <br> **Pulses** <br> (beans, peas, lentils, chickpeas) | Regular intake |
| **Fruits** (apples, pears, oranges, strawberries, blackberries, blueberries, pomegranate, blackberries, tangerines) <br> **Nuts** (walnuts, hazelnuts, pistachios) <br> **Seeds** (flax seeds, pumpkin seeds, sunflower seeds) <br> **Vegetables** (cucumber, tomatoes, onions, lettuce, beets, spinach, cabbage, broccoli, pumpkin, eggplant, peppers, carrots, mushrooms) <br> **Unrefined cereals and products** (wholegrain breakfast cereals, wholegrain toasted bread, brown rice, wholegrain pasta) | Regular intake |
| **Red wine** <br> **For flavor**: reduce salt and use more fresh herbs, garlic, citrus, and spices. | Low consumption |

## 9. Limitations and Future Directions

We believe that our study has proposed some useful recommendations, based on the literature, which could potentially assist with the alleviation of anxiety and depression through exercise and diet. However, it is necessary to highlight some limitations of the studies reported in this review.

Firstly, the underlying mechanisms of the benefits of exercise in people with depression is still unknown, so we could only hypothesize the involvement of TRP metabolism based on our review.

Secondly, other factors may be associated with the positive effect of MEN, such as brain derived neurotrophic factors (BDNFs) or lactate [114]. Therefore, an integrative pathway that considers these factors, which could affect well-being, should also be examined further.

Thirdly, the type of training applied, and the duration of the protocol are important factors to consider [14,115]. There are conflicting opinions about aerobic and anaerobic training, and which one has a better effect. Most of the literature described the effect of aerobic training on psychological disorders, however there is also considerable evidence that has highlighted the effect of anaerobic training on the CNS. For example, a recent study by Amato et al. demonstrated the positive benefits of anaerobic training in pathological subjects, increasing also BDNF levels and inducing improvements at a neurobiological level [116].

Future studies should address these factors and confirm the theoretical biochemical pathway suggested in our paper. This could occur through the evaluation of some biomarker levels pre- and post-activity, to elucidate whether MEN practices can elicit benefits at psychobiological level (Figure 2). It would also be prudent to conduct a study that evaluates the effects of the suggested protocol not only through blood chemistry, but also through psychological tests.

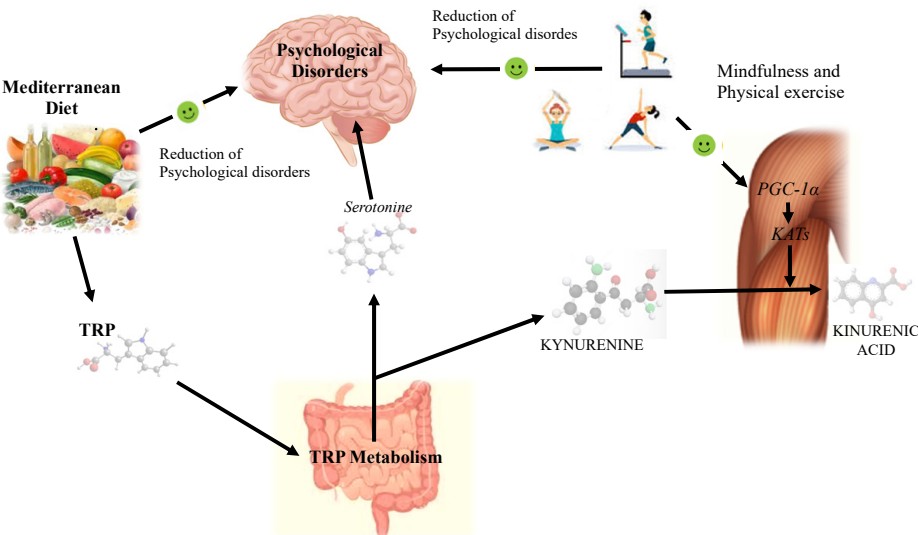

**Figure 2.** The hypothetical model of crosstalk MEN, "Mindfulness–Exercise–Nutrition".

**Author Contributions:** Conceptualization and writing, P.P., A.A. (Alessandra Amato) and S.B.; original draft preparation, P.P. and A.A. (Anna Alioto); review and editing, P.P., A.A. (Anna Alioto), A.A. (Alessandra Amato), G.M., M.R.B., R.S., S.B. and C.R. All authors have contributed to the article. All authors have read and agreed to the published version of the manuscript.

**Funding:** This research did not receive any external funding.

**Institutional Review Board Statement:** Not applicable.

**Informed Consent Statement:** Not applicable.

**Conflicts of Interest:** The authors declare no conflict of interest.

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
