# Peer review of "Fighting the Consequences of the COVID-19 Pandemic: Mindfulness, Exercise, and Nutrition Practices to Reduce Eating Disorders and Promote Sustainability"

_sustainability, doi:10.3390/su15032120_

Round 1

Reviewer 1 Report

Dear Authors.

The following is a review of the article entitled "Mindfulness, Exercise and Nutrition Practices on tryptophan’s shunt and environmental impact: How to achieve a sound mind in a sound body" which the aims to correlate the positive effect of mindfulness, aerobic exercise and diet on psychological disorders. Physical and mental wellbeing may be promoted not only for individuals with psychological disorders such as anxiety and depression, but also for those who have suffered stress as a result of the global pandemic. Thank you very much for thinking of me as a reviewer for this study.

After carefully reading the manuscript, I set forth comments and suggestions for the authors:

Title: Not appropriate. It does not specify objective, method or sample.

Abstract: No explain de aims of the study. The methodology is not properly explained. The results are not remarkable. The conclusions could be expanded. Add more practical application and some of the limitations found.

Keywords: Several keywords are repeated in the title.

Introduction:

It does not develop a theoretical framework on the study variables mindfulness, exercise and nutrition practices, psychological disorders, anxiety and depression, psychobiological biomarkers, Mediterranean diet and environmental sustainability More information, concepts, and more relevant studies.

The acronym "M.E.N." is incorrectly used.

Materials and Methods: It is recommended to use the PRISMA methodology (Preferred Reporting Items for Systematic reviews and Meta-Analyses), published in 2009, designed to help authors of systematic reviews to transparently document why the review was conducted, what the authors did and what they found. In addition, it is recommended to complete the analysis with the STROBE Statement, which is a reporting guide that includes a checklist of 22 elements that are considered essential for good reporting of observational studies.

Results:

The selected studies are not presented. No table or figure showing the results of the research. Tables and figures in the discussion section.

No appa format: Table 1. Information on typical foods of the Mediterranean diet and their frequency of consumption.

Discussion: All figures should be in the results section. The discussion should focus on the comparison of the results obtained with the results of other studies.

Conclusions: Added practical application. Irrelevant Figure 2. The hypothetical crosstalk model M.E.N., "Mindfulness-Exercise-Nutrition".

References: Corrects but some errors found in the references. Please check them.

Author Response

We would like to thank very much the referee for the time dedicated to the revision of our manuscript.

Title: as you suggested the title was changed to make one more appropriate. The new title is: Fighting the consequences of COVID-19 pandemic: Mindful-ness, Exercise and Nutrition Practices to reduce eating disorder and promote sustainability

Abstract: As you suggested we have rewritten the whole abstract in order to better point out what is the aim of this review, what we are discussing about and why it can be useful for the scientific community. Thanks for your suggestion and we hope you would like this revision.

Keywords: As you suggested the keywords were changed. Now they are: tryptophan’s shunt, malnutrition, obesity, plant-based diet, Mediterranean diet, homeostasis.

Introduction has been revised to focus directly on the aim of the study. A new chapter about Impact of Nutrition in systemic homeostasis and human well-being has been inserted.

The acronym "M.E.N." was changed to MEN.

Materials and Methods: Thank you for your comments. We did a narrative review so we did not use PRISMA as a basis because the nature of the review. To make more clear the point it was inserted the explanation in the material and method section (lines 63-69) We did not use STROBE Statement but we thank very much the referee for the information because we are leaning about it and we will use it as a useful tool for the future. Thank you very much.

Results and Discussion: We did not develop an individual  results session because we reviewed different aspect associated to the MEN practice. Therefore, we decided to insert tables and figures in the chapter to summarize new aspect associated to MEN and to discuss it.

Figure 2 has been changed with a new picture to be more readable.

References: done

Reviewer 2 Report

The manuscript is focused on the original topic and might be of interest to readers of the journal “Sustainability”. I have analyzed the article submitted for review with great care. The goal of this narrative review was  to summarize the empirical evidence of  the positive effect of mindfulness, aerobic exercise and diet on psychological disorders. The data sources used in this non-systematic review  were MEDLINE (PubMed), ScienceDirect, Web of Science, Scopus, and Google Scholar. Finally, 89 full-text articles were included to this analysis. In my opinion, the methodology for investigating did not raise any objections. The studies selection and eligibility screening are conducted properly. However, the dynamics of narrative review writing can be improve.
I overall have a generally good impression on this manuscript. The sections of the manuscript are well formulated and easy to follow. However, I would like to point out several issues.
1.    In the introduction section authors defined “the Mediterranean diet”, but the definitions of mindfulness and aerobic exercise are missed.
2.    In discussion authors provide subjective perspectives on the curried  topic. However, in my opinion, this section should be a bit more extensive.
3.    The authors stated that “Physical exercise increases the synthesis of the transcriptional coactivator PGC-1α1 in skeletal muscle, controlling many of the adaptations to physical activity and inducing the expression of KATs enzymes. In turn, this shifts peripheral kynurenine to kynurenic acid, and counteracts the toxic effects of kynurenine accumulation. This, added to the increase of TPH activity induced by physical activity, increases production of 5-HT and consequently serotonin” – What do you mean as a physical exercise?  Were here the aerobic or anaerobic exercise training effects ? 
4.    Please discuss the generalisability (external validity) of the study recommendation over a period of 24 weeks
5.    The quality (resolution) of Figure 2 should be improved

Author Response

Thank you very much for your positive comments about  the manuscript

 1)In the introduction section authors defined “the Mediterranean diet”, but the definitions of mindfulness and aerobic exercise are missed.

Thank you for your suggestion. You are right. In this new version we are dedicated an entire section to the “Mindfulness andPhysical Activity” in which we extensively discussed about that.

2) In discussion authors provide subjective perspectives on the curried topic. However, in my opinion, this section should be a bit more extensive.

Thank you for your suggestion. We have reviewed the section according to your suggestion.

 3) The authors stated that “Physical exercise increases the synthesis of the transcriptional coactivator PGC-1α1 in skeletal muscle, controlling many of the adaptations to physical activity and inducing the expression of KATs enzymes. In turn, this shifts peripheral kynurenine to kynurenic acid, and counteracts the toxic effects of kynurenine accumulation. This, added to the increase of TPH activity induced by physical activity, increases production of 5-HT and consequently serotonin” – What do you mean as a physical exercise?  Were here the aerobic or anaerobic exercise training effects ? 

Yes, we discussed about the effects of aerobic and anaerobic training.

4) Please discuss the generalisability (external validity) of the study recommendation over a period of 24 weeks

Following your comment, we underline the results of the main studies in this field and we added in Mindfulness and Physical Activity” section and change the time of intervention in the discussion’s section.

5) The quality (resolution) of Figure 2 should be improved

We changed the figure 2

Reviewer 3 Report

Please be consistent with word well-being or wellbeing. You change it throughout manuscript.

In latter part of Introduction paragraph

A Meditteranean type diet practiced world wide has been shown to prevent the development of coronary artery disease, diabetes and metabolic syndrome.  

The very next sentence on inverse relation would re word it for better flow.  

Kudos on this topic.  Interesting and should be helpful to those who are still suffering with anxiety and depression post Covid.  I liked the bullet point suggestions in the discussion.  I look forward to seeing this in print.  

Author Response

We would like to thank very much the referee for the time dedicated to the revision of our manuscript.

  1. Please be consistent with word well-being or wellbeing. You change it throughout manuscript.

         We changed the word well-being in all manuscript

  1. A Meditteranean type diet practiced world wide has been shown to prevent the development of coronary artery disease, diabetes and metabolic syndrome.  The very next sentence on inverse relation would re word it for better flow.  

         We rewrote the entire sentence (lines 306-312)

  1. Kudos on this topic.  Interesting and should be helpful to those who are still suffering with anxiety and depression post Covid.  I liked the bullet point suggestions in the discussion.  I look forward to seeing this in print.  

          Thank you very much for your compliments: we tried our best to help          the  suffering people of anxiety and depression post Covid.

Reviewer 4 Report

Dear Authors:

Thanks for this interesting article. The combination of mindfulness (or yoga, tai-chi, etc), physcial exercise and healthy eating styles is very important in order to tackle health problems related to our current lifestyles.

The article is well written, but in several sections maybe some facts are repeated redundantly. Please correct this (i.e. MD - definition appears several times in the text).

Please use also consistently the abbreviations from the first moment on (except abstract) (i.e. MD, TRP, etc).

I would also suggest to include lines on every page and please number pages also in order to make the review easier for us and for you when replying to our comments and suggestions. 

Please use also the MDPI guidlines for references and [...] instead of (...).

What do you mean when mentionning MD ... intake of fish and fruits seafood ?  Shouldn't it be seafood and fruits ? 

Last paragraph of introduction: 

I think it would be godd to add some sentence on the importance of M.E.N. for even healthy persons, not only suffering from stress or having other disorders.

Materials and Methods:

I agree that narrative reviews or rapid reviews are more and more employed in scientific articles. But, this kind of methodology has also its flaws and pros. Could you please add something on this? So that the reader could consider the narrative review as a valuable methodology in comparison with other systematic reviews.

M.E.N.:

IS MEN as a unit discussed in your article? as a coherent and comprehensive tool ? 

Some minor errors / mistakes in english need to reviewed (i.e. the WHO have (has!) reported... )

MD:

- Please use consistenly the abbreviation of MD in the text.

- Too often MD is defined in the text. It would be enough to do so at the beginning when MD is first mentionned.

- You are citing ref [78] Keys. A. What is the situation nowadays? this study dates from 1970.

- Low environmental impacts of MD ? Why do think this is still the case? What about the aggressive and massive agriculture in Spain?

Discussion:

The HRmax heart rate for exercising is estimated by the Karvonen et al. (1957) formula! There are more recent and accurate formulas in the "market".

Is table 1 really necessary for the comprehension of the article?

Limitatons and future directions:

Figure 2. Please make Figure 2 more readable. Precise and higher resolution.

Is this really the best place for this  figure? Maybe it would be better to show it at the end of the introduction section.

Author Response

We would like to thank very much the referee for the time dedicated to the revision of our manuscript.

Dear Authors:Thanks for this interesting article. The combination of mindfulness (or yoga, tai-chi, etc), physcial exercise and healthy eating styles is very important in order to tackle health problems related to our current lifestyles.

Thank you very much for your comments.

The article is well written, but in several sections maybe some facts are repeated redundantly. Please correct this (i.e. MD - definition appears several times in the text).

Please use also consistently the abbreviations from the first moment on (except abstract) (i.e. MD, TRP, etc).

done

I would also suggest to include lines on every page and please number pages also in order to make the review easier for us and for you when replying to our comments and suggestions. 

done

Please use also the MDPI guidlines for references and [...] instead of (...)

done

What do you mean when mentionning MD ... intake of fish and fruits seafood ?  Shouldn't it be seafood and fruits ? 

Yes, you are right. The sentence has been changed in seafood and fruits. Please see lines 132-133

Last paragraph of introduction:  

I think it would be godd to add some sentence on the importance of M.E.N. for even healthy persons, not only suffering from stress or having other disorders.

Yes, you are right. We have remodeled the introduction’s section and we added new sentences about the impact of the MEN in healthy subjects at the end of the new section 2 renamed Impact of Nutrition in systemic homeostasis and human well being. Please see lines 137-144

Materials and Methods:

I agree that narrative reviews or rapid reviews are more and more employed in scientific articles. But, this kind of methodology has also its flaws and pros. Could you please add something on this? So that the reader could consider the narrative review as a valuable methodology in comparison with other systematic reviews.

Thank you for your suggestion. We added a small paragraph to better explain the relevance of the narrative review into the material and methods section

M.E.N.: 

IS MEN as a unit discussed in your article? as a coherent and comprehensive tool ? 

M.E.N was intended as an acronym. To make it more clear we changed the acronym "M.E.N." to MEN.

Some minor errors / mistakes in english need to reviewed (i.e. the WHO have (has!) reported... )

done. Thanks

Please use consistenly the abbreviation of MD in the text.

done

Too often MD is defined in the text. It would be enough to do so at the beginning when MD is first mentionned.

done

You are citing ref [78] Keys. A. What is the situation nowadays? this study dates from 1970.

We indicated this reference taking into consideration the historical period of the 50's

Low environmental impacts of MD ? Why do think this is still the case? What about the aggressive and massive agriculture in Spain?

We have added some sentence and the reference about the aggressiveness of the Spanish agriculture.

Discussion

The HRmax heart rate for exercising is estimated by the Karvonen et al. (1957) formula! There are more recent and accurate formulas in the "market".

Yes is true that there are other formulas in the market but as you can see in the paper from Ignaszewski, M., et al 2017 (Ignaszewski, M., Lau, B., Wong, S., & Isserow, S. (2017). The science of exercise prescription: Martti Karvonen and his contributions. British Columbia Medical Journal, 59(1), 38-41) Karvonen's formula is the most used so far.

Is table 1 really necessary for the comprehension of the article

Thanks for suggestion. We think that it can be useful to reach large audience.

Limitatons and future directions:

Figure 2. Please make Figure 2 more readable. Precise and higher resolution.

Thanks. Done.

Is this really the best place for this  figure? Maybe it would be better to show it at the end of the introduction section

The figure at the end of the manuscript summarized the topic. We would prefer to leave it there but if you do not agree we will move it where you think is more appropriate. Thanks again for the time you have dedicated to the revision of the manuscript.

Round 2

Reviewer 1 Report

Dear Authors.

After the second revision of the article entitled "Fighting the consequences of COVID-19 pandemic: Mindfulness, Exercise and Nutrition Practices to reduce eating disorder and promote sustainability" the authors have improved the points indicated. The manuscript is considered ready for publication.

Regards

Reviewer 4 Report

Dear Authors. Thank you very much for correcting all the comments and errors. 

Best wishes.